# Pathogenesis of Oral Toxicities Associated with Targeted Therapy and Immunotherapy

**DOI:** 10.3390/ijms24098188

**Published:** 2023-05-03

**Authors:** Alessandro Villa, Michal Kuten-Shorrer

**Affiliations:** 1Oral Medicine, Oral Oncology and Dentistry, Miami Cancer Institute, Baptist Health South Florida, Miami, FL 33176, USA; 2The Herbert Wertheim College of Medicine, Florida International University, Miami, FL 33176, USA; 3Department of Orofacial Sciences, University of California San Francisco, San Francisco, CA 94143, USA; 4Eastman Institute for Oral Health, University of Rochester Medical Center, Rochester, New York, NY 14642, USA

**Keywords:** targeted therapies, immunotherapy, immune checkpoint inhibitors, stomatitis, oral mucositis, oral toxicity

## Abstract

Targeted therapy and immunotherapy have redefined cancer treatment. While they have enhanced tumor response and improved survival rates in many cancer types, toxicities continue to occur, and these often involve the oral cavity. Broadly reported as “mucositis” or “stomatitis,” oral toxicities induced by targeted therapies differ clinically and mechanistically from those associated with conventional chemotherapy. Manifesting primarily as mucosal lesions, salivary gland hypofunction, or orofacial neuropathies, these oral toxicities may nonetheless lead to significant morbidity and impact patients’ quality of life, thereby compromising clinical outcomes. We conclude that familiarity with the spectrum of associated toxicities and understanding of their pathogenesis represent important areas of clinical research and may lead to better characterization, prevention, and management of these adverse events.

## 1. Introduction

Targeted therapies, and more recently immunotherapy, have dramatically changed the landscape of treatments of a variety of cancers and improved clinical outcomes [1]. Unlike conventional cancer treatment regimens, new agents interfere with the growth and survival of cancer cells by interacting with the specific receptors and intracellular signaling pathways involved in the carcinogenesis and cancer progression [1]. These treatments include anti-tumor monoclonal antibodies (mAbs), small molecules, signal transduction receptor inhibitors, and cancer vaccines [2].

The global market for targeted therapies, continues to expand, and the growth of this market is mainly attributed to the rising prevalence of target diseases, the increasing demand for targeted agents, the increasing adoption of immunotherapy drugs over conventional treatments, and a favorable approval scenario.

Targeted cancer therapies are often used as first and second line treatments for several solid tumors, including those of breast, colorectal, head and neck, lung, and pancreatic cancers, as well as hematologic malignancies such as lymphoma, leukemia, and multiple myeloma [2]. The benefit from targeted therapy and immune checkpoint inhibitors is however tempered by toxicities that affect different sites, including the oral cavity. Agents frequently reported to be associated with oral complications include tyrosine kinase inhibitors (TKIs), inhibitors of the VEGF receptor (VEGFR), anti-epidermal growth factor receptor (anti-EGFR), vascular endothelial growth factor (VEGF), mammalian target of rapamycin inhibitors (mTORis), and, more recently, immune checkpoint inhibitors (ICIs) [3] (Table 1).

The clinical features of the toxicities of targeted therapy and immunotherapy differ from those described in patients receiving cytotoxic chemotherapy, and the severity largely depends on the type of medication considered, the dosage, and the indication of use [4,5,6]. Oral mucosal changes secondary to targeted therapies and immune checkpoint inhibitors are often reported as “mucositis” or “stomatitis” in clinical trials, although they frequently have a distinctive clinical presentation from the oral mucosal injuries associated with conventional cytotoxic drugs.

Several other oral complications from targeted therapy have been described, and these include infections, orofacial neuropathies (e.g., dysesthesia), dysgeusia, xerostomia, salivary gland hypofunction, and osteonecrosis of the jaw [3,7]. These oral complications can occur at any time during the patient’s treatment course, and may result in both acute and chronic toxicities, some of which may persist even after the discontinuation of the agent. Oral complications may lead to patient morbidity and negatively affect patients’ physical and psychological well-being [3,7]. Additionally, they can affect the cancer treatment dosing schedule, and they are associated with significant increased cost.

Oral toxicities of targeted therapy and immunotherapy require careful work up and management. Herein, we review the pathogenesis and clinical manifestations of oral toxicities from oncologic targeted therapies, including ICIs. For the purpose of this manuscript, we excluded bone-modifying agents indicated for bone metastasis and osteoporosis.

## 2. Oral Toxicities

### 2.1. Stomatitis

“Stomatitis” usually refers to oral inflammatory conditions, while the term “mucositis” refers to the mucosal damage secondary to radiation therapy and conventional chemotherapy [8]. The following section focuses on the pathobiology of stomatitis secondary to targeted therapy and immunotherapy agents.

#### 2.1.1. Stomatitis Associated with Mammalian Target of Rapamycin Inhibitors (Table 1)

Mammalian target of rapamycin inhibitors represent a group of targeted agents that are used for the treatment of several malignancies of the pancreas, lung, and advanced breast cancer, to prevent graft rejection in solid organ transplantation, and for the prophylaxis and management of graft-versus-host-disease in patients who receive allogeneic hematopoietic stem cell transplantation [9,10,11]. Everolimus, sirolimus, and temsirolimus are the mTORis currently approved by the Food and Drug Administration (FDA) in the Unites States. Both sirolimus and everolimus bind to the FK binding protein to moderate the mTOR activity and subsequently inhibit the PTEN/PI3K/Akt and JAK pathways [12]. The mTOR downregulates the interleukin (IL)-2-mediated signal transduction, which leads to a G1-S phase cell-cycle arrest. In addition, it blocks the response of T and B cell activation via IL-2 and IL-5, which prevents cell-cycle proliferation and progression.

Aphthous-like oral ulcers represent a common debilitating adverse event of mTORis. In 2010, Sonis et al. proposed the term “mTOR Inhibitor Associated Stomatitis (mIAS)” to differentiate the oral lesions secondary to mTORis from the oral ulcerations secondary to cytotoxic chemotherapy or radiotherapy to the head and neck (“conventional oral mucositis”) [13]. mIAS affects 25–55% of patients and manifests as round ulcers of the nonkeratinized mucosa covered by a grayish-yellow fibrin pseudomembrane surrounded by erythema [14,15,16]. The risk of mIAS depends on different genetic factors [17]. Management is typically carried out with topical steroid agents, although severe cases may require systemic steroid therapy or the discontinuation of the mTORi. mIAS is a dose-limiting oral adverse event.

The pathogenesis of mIAS remains poorly understood, although it appears to share similar pathobiological pathways with recurrent aphthous stomatitis (RAS), a common idiopathic immune-mediated oral disease [18]. It has been hypothesized that mTORis have a direct effect on the oral epithelium. In an animal study, Mills et al. showed that sirolimus exposure led to a delay in wound repair in mice due to defects in T cell proliferation and function and the production of growth factors [19]. Sonis et al. have suggested a three-phase sequence for the development of mIAS: (1) a direct epithelial injury secondary to the exposure to an mTORi; (2) the release of pro-inflammatory cytokines; and (3) the activation of the innate immune system response with an autoimmune-like inflammatory response and the subsequent entry of acute inflammatory cells [20]. In particular, in their organotypic model, everolimus was shown to inhibit the proliferation of epithelial cells and induce apoptotic changes with the formation of intra-epithelial vacuoles. Histologically, the organization and integrity of the epithelium was also compromised following exposure to an mTORi. When the cytokine levels were considered, IL-6 and IL-8 were significantly higher in everolimus-treated tissue compared with healthy tissue. However, no differences were observed for IFN-c or any of the other cytokines that are increased in RAS. The increase in the levels of IL-1a, b and TNF-a in everolimus-treated supernatant was attributed to changes to the epithelium because of culturing and unrelated to everolimus. Finally, mTORis may also bind and inhibit other inflammatory mediators and angiogenesis mediators, including nitric oxide and VEGF [21].

#### 2.1.2. Stomatitis Secondary to Anti-EGFR Agents and VEGFR Inhibitors (Table 1)

EGFR inhibitors (EGFRIs) were among the first targeted therapies developed for the treatment of epithelial tumors, and they are used to treat advanced/metastatic NSCLC, pancreatic cancer, breast cancer, colorectal cancer, and head and neck (H&N) cancer. EGFRIs can be divided into two classes: mAbs (e.g., cetuximab, panitumumab, trastuzumab) and small-molecule TKIs (e.g., gefitinib, erlotinib). As wild-type EGFR plays a critical role in homeostatic regulation in epidermal and epithelial cells, most agents targeting EGFRs produce a similar spectrum of mucocutaneous toxicities [22,23]. Oral mucosal lesions associated with EGFRIs, often reported as “stomatitis” or “mucositis,” present as moderate erythema with limited, well-defined, and superficial ulcers primarily involving the nonkeratinized mucosa [22]. Compared with anti-EGFR mAbs monotherapy, a significantly greater risk of developing oral mucosal lesions is seen with EGFR TKIs such as erlotinib, afatinib, and dacomitinib. When combined with conventional chemotherapy, however, anti-EGFR mAbs such as cetuximab and panitumumab may increase the risk and severity of mucosal involvement, with a combined presentation of both superficial and deeper, classic oral mucositis ulcers [24,25]. Stomatitis is also reported with VEGFR-directed multitargeted tyrosine kinase inhibitors (TKIs) such as sunitinib, sorafenib, pazopanib, and cabozantinib. While in this case the broad term “stomatitis” most frequently encompasses reports of dysesthesia and dysgeusia (see Section 2.3), oral ulceration may also occur, presenting as discrete linear ulcers of the nonkeratinized mucosa [7,26]. The management of oral mucosal lesions caused by EGFRIs and multitargeted TKIs follows the expert recommendations for targeted therapy-associated stomatitis published by the European Society of Medical Oncology [27]. As with the management of mIAS, in addition to basic oral care and oral hygiene recommendations, the use of high-potency steroids (topical, intralesional, or systemic) is recommended as a first-line therapy.

### 2.2. Red and White Lesions

Oral mucosal lesions are a potential side effect of both targeted therapy and immunotherapy, with their severity and clinical presentation varying depending on the agent used. Oral lesions can range from mild hyperkeratotic changes to severe ulcerations, and their presence can significantly impact a patient’s quality of life. In this section, we will review the pathobiology of oral mucosal lesions associated with ICIs and targeted agents.

#### 2.2.1. Oral Mucosal Immune-Related Adverse Events (Table 1)

The use of ICI therapy has exponentially increased over the past few years and revolutionized the treatment of several hematologic malignancies and solid organ cancers. ICIs target programmed cell death-1 (PD-1), PD ligand-1 (PD-L1), and cytotoxic T-lymphocyte-associated antigen-4 (CTLA-4), with a subsequent disruption of the host T cell signaling and the upregulation of the T cell immune innate and adaptive response against cancer cells [28]. The inhibition of PD-1, PD-L1, and CTLA-4 can lead to autoinflammatory and autoimmune responses that affect several organs, including the oral cavity [29]. Oral toxicities secondary to CTLA-4 and PD-1/PD-L1 have been reported in approximately 8% of patients and are often associated with other non-oral immunotherapy-related adverse events (irAEs) [30,31]. However, the real prevalence of oral irAEs remains poorly understood due inconsistencies in toxicity reporting [30,31,32].

Oral mucosal lesions may resemble oral lichen planus and mucous membrane pemphigoid/bullous pemphigoid; some cases are characterized by large ulcerations and crusting of the lips, similar to what it is observed in patients with erythema multiforme and Stevens–Johnson syndrome/toxic epidermal necrolysis [33,34,35]. Treatment of oral mucosal irAEs is carried out with high-potency topical steroids, although some patients may benefit from systemic steroid therapy or steroid-sparing immunosuppressive agents [36,37]. Severe cases may require holding ICI therapy until the mucosal lesions improve or resolve [34,36].

As with cutaneous toxicities, the majority of mucosal oral irAEs seem to be secondary to the activation of adaptive immunity, with most cases reported in patients receiving PD-1/PD-L1 inhibitors. Studies have shown that PD-1 inhibitors induce increased CD8^+^ T cells in the tissue, and the inhibition of CTLA-4 leads to increased CD4^+^ T cells in the lymph nodes [37,38]. The expansion of T cell receptor (TCR) diversity, increased detection of autoantibodies during the first 30 days of ICI therapy, and T cell and B cell clonality have been associated with the development of irAEs [29]. Interestingly, the use of TNF-α (e.g., infliximab) and IL-6 (e.g., tocilizumab) inhibitors as steroid-sparing agents has been recommended for the management of several irAEs, suggesting that cytokine levels (general or tissue-specific) may also play a role in the pathogenesis of irAEs [39,40]. Indeed, both CTLA-4 and PD-1/PD-L1 inhibition result in the increased production of cytokines, including TNF, INF-γ, and IL-2, which can lead to further T cell proliferation and activation [41]. Further study is required to elucidate the precise role these cytokines play in the development of irAEs [42].

#### 2.2.2. Lichenoid Lesions Associated with Imatinib (Table 1)

Imatinib mesylate is a first-generation BCR-ABL inhibitor which also targets the platelet-derived growth factor receptor (PDGFR) and C-KIT kinases. Since its initial approval by the FDA in 2001 for the treatment of chronic myelogenous leukemia (CML), imatinib has become the standard of care in CML and gastrointestinal stromal tumors (GIST), and its use has been extended to various other malignant and hematological disorders. Oral lichenoid reactions are the most frequent imatinib-associated oral adverse event, occurring with or without cutaneous involvement [22,43,44]. The lichenoid lesions are typically asymptomatic, presenting with characteristic reticular, erosive, and/or ulcerative features. Symptomatic cases can be managed with high potency topical corticosteroid therapy, and imatinib treatment can be continued without interruption in most cases. The primary objective of the treatment is to alleviate discomfort as well as to reduce or resolve erythema and ulcers. In some cases, systemic corticosteroids, as well as other immunosuppressive drugs, may be required.

#### 2.2.3. Rituximab and Oral Lichenoid-like Lesions (Table 1)

Oral lichenoid reactions have also been anecdotally reported in association with rituximab [45,46], an anti-CD20 mAb indicated for the treatment of the majority of B cell non-Hodgkin lymphomas (NHLs). As with the management of oral mucosal irAEs, the management of targeted therapy-induced lichenoid lesions is focused on symptom control. High-potency topical steroids are the mainstay of treatment, yet it may be necessary to escalate treatment to the use of systemic therapies in severe cases [22].

#### 2.2.4. Hyperkeratotic Lesions and Possible Increased Risk of Squamous Cell Carcinoma (SCC) (Table 1)

Various malignancies are driven by aberrations in the RAS-RAF-MEK-ERK mitogen-activated protein kinase (MAPK) signaling pathway, which regulates cellular proliferation, differentiation, and survival. The most common mutation leading to the overactivation of the MAPK pathway is the BRAF^V600E^ mutation, found in up to 60% of cases of malignant melanoma. The BRAF inhibitors (BRAFIs) vemurafenib, dabrafenib, and encorafenib have revolutionized the treatment of BRAF^V600^-mutant melanoma, resulting in meaningfully improved progression-free survival. In addition to melanomas, BRAFIs have been approved to treat advanced forms of other BRAF^V600E^-mutated cancers, such as thyroid, colorectal, and NSCLC [47]. Dermatological AEs are the most significant and frequent toxicity associated with BRAFIs, particularly induced hyperkeratotic lesions, ranging from benign verrucous keratoses to invasive squamous cell carcinoma (SCC) [23]. These are thought to be the result of the paradoxical activation of the MAPK pathway in wild-type BRAF keratinocytes and subsequent keratinocyte hyperproliferation. Indeed, combination therapy with a BRAFI and a MEK inhibitor (vemurafenib–cobimetinib, dabrafenib–trametinib, or encorafenib and binimetinib) is associated with reduced dermatological toxicity because it blocks the MAPK pathway downstream [48]. While the skin is primarily involved, oral mucosal hyperkeratosis has also been reported to occur within the first weeks of treatment, and there has been a single report of SCC of the labial mucosa developing in a patient treated with vemurafenib [22,49]. Lesions can be found on both the keratinized and non-keratinized mucosa, including the buccal mucosa, gingival margin, and hard palate. With no specific management for these lesions, routine oral examinations and biopsy of keratotic lesions is recommended.

#### 2.2.5. Geographic Tongue (Table 1)

Cases of geographic tongue, or benign migratory glossitis, have been reported to occur in association with antiangiogenic targeted therapies, including the anti-VEGF mAbs, bevacizumab, and the nonselective multitargeted TKIs sorafenib, sunitinib, axitinib, etc. [50,51,52]. Clinically, it is characterized by erythematous lesions with filiform papillae atrophy, surrounded by a white peripheral rim of the tongue. Geographic tongue secondary to antiangiogenic targeted therapies and nonselective multitargeted TKIs is often asymptomatic. Some cases can be associated with pain and/or a burning sensation, but do not usually require any treatment modification. The use of topical anesthetic agents, diphenhydramine solution as a swish and spit, or topical corticosteroids can be considered for symptomatic cases. It is of note that the existing literature does not clearly establish whether patients develop geographic tongue as a result of targeted therapy, whether the condition worsens following the administration of the targeted agents, or whether the lesions are already present prior to starting therapy. Additional research is needed to establish the precise nature of this possible association.

The pathogenesis of geographic tongue secondary to targeted therapy is not well understood. Hubiche et al. have suggested that since VEGF or VEGF receptors play a crucial role in maintaining the homeostasis of the buccal mucosa, as well as in certain oral diseases, angiogenesis inhibitors for targeting these specific molecules could potentially induce the development of geographic tongue [52].

### 2.3. Oral Dysesthesia

Oral dysesthesia is defined as an abnormal sensation in the oral cavity, such as a burning feeling, that is not associated with any abnormal clinical findings [53]. It is often described as a subjective feeling of oral tingling or pain, and it can be a diagnostic challenge for many clinicians. Several targeted agents have been associated with this complication.

#### Oral Dysesthesia of Multitargeted Tyrosine Kinase Inhibitors (Table 1)

Multitargeted TKIs are a class of antineoplastic drugs which offer a target-specific approach to antitumor therapy [54,55,56]. In particular, multitargeted TKIs block several molecular targets, including VEGFR-2, VEGFR-3, Flt-3, C-Kit, the PDGF receptor, and c-Raf and b-Raf kinases. As a consequence, proliferative pathways are blocked, reversing the mechanism of tumor nutrition and growth and increasing tumor regression by inhibiting cell survival [54,55].

Oral toxicity secondary to multitargeted TKI therapy is a common AE reported by 20–35% of patients [57,58,59]. This toxicity has often been described in clinical trials as “stomatitis” or “mucositis” [56,60,61,62,63]. However, patients usually complain of an oral burning sensation or sensitivity with or without dysgeusia and xerostomia, and with a normal appearing oral mucosa [26]. Therefore, the term “oral dysesthesia” has been recommended to better characterize this condition [59]. Oral dysesthesia is often seen with VEGFR-directed multi-target TKIs, such as sunitinib, sorafenib, pazopanib, and cabozantinib, developing at a median of 0.5–1.4 months after the beginning of treatment [26]. Management focuses on relieving symptoms and is similar to the recommended approach for patients with burning mouth syndrome. It involves using topical benzodiazepines or systemic anti-convulsant drugs and tricyclic anti-depressants, with varying responses [26,64]. The pathogenesis of TKI-induced oral dysesthesia remains poorly understood.

### 2.4. Dysgeusia

Dysgeusia (taste alterations) is a common adverse effect of cancer treatment and may be explained by neurological or mucosal damage. With targeted therapies, dysgeusia may be mechanistically related to target inhibition in the normal tissue.

#### 2.4.1. Dysgeusia Associated with Hedgehog Signaling Pathway Inhibitors (Table 1)

Hedgehog signaling pathway (HhSP) inhibitors (e.g., vismodegib and sonidegib) are a class of drugs used in the treatment of advanced, metastatic, or unresectable basal cell carcinoma (BCC). HhSP has been found to regulate the differentiation and maintenance of lingual taste receptor cells (such as the lingual papillae and taste buds), thereby playing a critical role in taste function integrity [65]. It is therefore not surprising that taste alterations are among the most common AEs associated with HhSP inhibitors, reported to occur in 55.8% and 44.3% of vismodegib and sonidegib-treated patients, respectively [66]. Kumari et al. investigated the role of the Hh pathway in taste sensation by using sonidegib (LDE225) to inhibit signaling at Smoothened, a key regulator of Hh signaling [67]. Normally, when the Hh ligand binds to the membrane receptor Patched, Smoothened is activated and initiates a signaling cascade that leads to the activation of Gli transcription factors. Treatment with sonidegib demonstrated a direct and essential requirement of the Hh pathway in maintaining taste bud function and homeostasis. Furthermore, the results of this study, along with other studies using the HPI drug sonidegib, showed that disrupting Hh signaling led to a rapid loss of taste buds and reduced levels of Shh ligand within the taste buds, indicating a loss of taste bud cells.

Typically developing early in the course of treatment, HhSP inhibitor-associated dysgeusia is primarily mild to moderate in severity (grades 1–2 using the Common Terminology Criteria for Adverse Events) and reversible, with most cases resolving within six months of vismodegib discontinuation [68]. Nonetheless, due to the high incidence of HhSP-induced taste changes, it is necessary to inform and educate patients about this potential AE at the initiation of therapy [22,69]. Early nutritional screening followed by routine counseling from a dietician should be considered in order to prevent significant weight loss and nutritional compromise, and to mitigate the risk of subsequent treatment interruption.

#### 2.4.2. Multitargeted TKI-Associated Dysgeusia (Table 1)

Dysgeusia is also a common toxicity reported with crizotinib, a multitargeted TKI of anaplastic lymphoma kinase (ALK), mesenchymal-to-epithelial transition (MET) protein, and ROS proto-oncogene 1 (ROS1) used for the treatment of ALK-positive non-small cell lung cancer (NSCLC). While dysgeusia has also been reported with the next-generation ALK inhibitor alectinib, the incidence was found to be lower, and, in one case, a grade 3 crizotinib-induced dysgeusia was successfully treated by switching to alectinib [70,71]. A possible explanation is alectinib’s high selectivity for ALK without activity against MET and ROS1, though evidence for MET and/or ROS1 involvement in taste function is lacking.

#### 2.4.3. Dysgeusia Secondary to Immune-Checkpoint Inhibitors (Table 1)

Neurologic AEs have been reported in 1–12% of patients undergoing immunotherapy [72]. Peripheral neuropathy is rare and may manifest in the oral cavity as dysgeusia and oral dysesthesia. A recent meta-analysis showed that although dysgeusia has been reported in patient receiving ICI therapy, the risk is lower compared with conventional chemotherapy regimens [73]. The damage to peripheral nerves in neurologic irAEs appears to be secondary to cell-mediated mechanisms, antibody responses to compact myelin, Schwann cells, or nodal antigens as part of an abnormal immune response [74]. In addition, cross-reactivity between the tumor antigens and similar epitopes on healthy cells has been reported as another possible mechanism of the neurologic toxicity of ICIs [75].

#### 2.4.4. Other Targeted Agents and Dysgeusia

Cases of dysgeusia accompanied by xerostomia have also been reported in patients receiving everolimus and temsirolimus, although the underlying mechanisms have not yet been determined [76].

### 2.5. Salivary Gland Hypofunction (Table 1)

Patients with salivary gland hypofunction may complain of severe dry mouth with Sjögren syndrome-like clinical features and symptoms. However, most cases of dry mouth secondary to ICI therapy seem to be mediated mainly by autoreactive T cells and the T cell-mediated inflammation of salivary glands rather than the B cells typical of Sjögren syndrome [77]. Indeed, labial salivary gland biopsies obtained from ICI-induced sicca patients demonstrated marked sialadenitis with increased CD3+T cell infiltration and acinar injury, but a virtual absence of CD20+ B cells. Furthermore, only a few ICI-induced patients were seropositive for anti-Sjögren syndrome-related antigens A or B (Anti-SSA/B) autoantibodies, which may have been pre-existing. The management of immunotherapy-related xerostomia (“subjective feeling of oral dryness”) and dry mouth is carried out with oral moisturizers or sialogogue therapy (pilocarpine and cevimeline) [36,78,79]. With the increased risk of dental sequelae (e.g., caries, recurrent candidiasis infections), regular dental examinations, including frequent dental prophylaxis and the prescription of topical fluoride treatments, is recommended.

### 2.6. Gingival Bleeding (Table 1)

Patients treated with antiangiogenic targeted therapies, VEGF inhibitors, and VEGFR inhibitors are also at increased risk of bleeding and delayed wound healing because of the effects of these therapies on vascular permeability and proliferation [80,81]. The precise mechanism by which bleeding is precipitated has not been full elucidated. The main hypothesis is that VEGF signaling plays a role in promoting endothelial cell survival and integrity in the adult vasculature, and that therefore its inhibition interferes with the regenerative capacity of damaged endothelial cells and causes capillary leakage. Mild, spontaneous mucosal bleeding has been reported in 20% to 40% of patients treated with bevacizumab. While epistaxis is the most frequently reported bleeding event, gingival bleeding has also been reported. Similar mucocutaneous bleeding events have also been associated with sunitinib and sorafenib [82]. Bleeding events and delayed wound healing should be taken into consideration before oral surgery.

### 2.7. Hyperpigmentation (Table 1)

Pigmentary changes are a well described AE of imatinib mesylate, with hypopigmentation being reported far more often than hyperpigmentation [83]. Intraorally, imatinib use is associated with a characteristic blue-gray, asymptomatic hyper-pigmentation of the hard palate, representing, histologically, depositions of melanin in the lamina propria [84]. Anecdotal cases of intraoral hyperpigmentation affecting other sites, such as the gingivae or teeth, have also been reported [85,86]. A recent multivariant analysis in a cross-sectional study of 74 participants found that the duration of imatinib therapy is directly proportional to the intensity and extent of the hyperpigmentation observed, especially in cases with hydroxyurea treatment preceding imatinib therapy [87]. The pathophysiology underlying the pigmentary changes relating to imatinib therapy, and specifically the intraoral hyperpigmentation, remains unclear. Among the implicated mechanisms are the direct inhibition of C-kit, which is physiologically expressed in the oral mucosa, and the deposition of drug metabolite complexes [88,89]. Imatinib is known to target the ATP-binding site of the Bcr-Abl tyrosine kinase, as well as other tyrosine kinases such as platelet-derived growth factor receptor-b, C-kit, and C-ABL [89]. C-kit is a transmembrane growth factor expressed in melanocytes, basal skin cells, and mast cells. Stimulation of C-kit leads to the activation of the microphthalmia transcription factor (MITF), which then transactivates the promoter of the tyrosinase pigmentation gene of melanocytes [83]. Imatinib is thought to inhibit ligand binding to specific receptors on the surface of human melanocytes, which reduces cellular activity and may cause hypopigmentation and, less frequently, hyperpigmentation of the skin and mucosa [90]. This may be due to a metabolite of the drug chelating iron and melanin, as with the action of other drugs (e.g., minocycline and chloroquine). The reason why the hard palatal mucosa is often affected is not yet understood, but it is known that the palatal mucosa contains a large number of mucosal melanocytes where imatinib metabolites can accumulate. Additionally, C-kit signaling may also play a role in oral hyperpigmentation as it is expressed in the mesenchymal cells of the human oral cavity, e.g., dental pulp cells and gingival fibroblasts.

As the hyperpigmented lesions are benign, no treatment is usually required. Laser therapy may be recommended for individuals with extensive pigmentations and/or aesthetic concerns.

## 3. Conclusions

Oral toxicities secondary to targeted therapy represent a unique challenge, and the understanding of the pathogenesis of many of these complications is still limited. Not only is the population of cancer patients and survivors growing, but there continue to be innovations in modern oncology with new targeted agents and treatments, many of which are associated with both acute and chronic oral adverse events. Oral toxicities may manifest as mucosal lesions (such as mIAS and oral immune-related adverse events), salivary gland hypofunction, and orofacial neuropathies (e.g., oral dysesthesia from multitargeted TKIs and dysgeusia secondary to HhSP inhibitors). The severity of these oral complications depends on the type of targeted agent considered, oral health-related factors, and comorbidities. Most toxicities require supportive care and are managed using a multi-disciplinary approach.

A better understanding of the pathogenesis of oral toxicities from targeted therapy may lead to the development of reliable biomarkers for AE prediction and monitoring as well as new prevention strategies and therapeutic options in the future.

## Figures and Tables

**Table 1 ijms-24-08188-t001:** Oral toxicities associated with targeted therapies and immunotherapies: summary table.

Oral Toxicity	Drug	Class	Main Treatment Suggestions
Stomatitis	Cetuximab, panitumumab, trastuzumab; gefitinib, erlotinib, afatinib, dacomitinib Sunitinib, sorafenib, pazopanib, cabozantinib	EGFR inhibitors (monoclonal antibodies; small-molecule tyrosine kinase inhibitors) Multitargeted tyrosine kinase inhibitors	Basic oral care, steroids (topical, intralesional, oral), analgesics.
mIAS	Everolimus, sirolimus, temsirolimus	mTOR inhibitors	Steroids (topical, intralesional, oral).
Lichenoid lesions	Imatinib Rituximab	BCR-ABL inhibitor Anti-CD20 monoclonal antibody	Steroids (topical, intralesional, oral).
Oral mucosal irAEs	Nivolumab, pembrolizumab, cemiplimab, ipilimumab, atezolizumab	Immune checkpoint inhibitors	Steroids (topical, intralesional, oral).
Hyperkeratotic lesions	Vemurafenib, dabrafenib, encorafenib	BRAF inhibitors	No specific interventions. Routine examination and biopsy in case of irregular lesions.
Geographic tongue	Bevacizumab Sunitinib, sorafenib, axitinib	Anti-VEGF monoclonal antibody Multitargeted tyrosine kinase inhibitors	Topical steroids for symptomatic cases.
Dysesthesia	Sunitinib, sorafenib, pazopanib, cabozantinib	Multitargeted tyrosine kinase inhibitors	Topical or systemic clonazepam, gabapentin, antidepressants.
Dysgeusia	Vismodegib, sonidegib Crizotinib Nivolumab, pembrolizumab, cemiplimab, ipilimumab, atezolizumab Everolimus, temsirolimus	HhSP inhibitors Multitargeted tyrosine kinase inhibitor Immune checkpoint inhibitors mTOR inhibitors	Consider consultation with dietitian. Taste usually improves after discontinuation of medication.
Immune related salivary gland hypofunction	Nivolumab, pembrolizumab, cemiplimab, ipilimumab, atezolizumab	Immune checkpoint inhibitors	Basic oral care, topical mucosal lubricants, saliva substitutes, masticatory stimulants (sugar-free chewing gum or lozenges), sialogogues (pilocarpine, civemiline).
Gingival bleeding	Bevacizumab Sunitinib, sorafenib	Anti-VEGF monoclonal antibody Multitargeted tyrosine kinase inhibitors	No specific interventions. Maintain oral hygiene.
Hyperpigmentation	Imatinib	BCR-ABL inhibitor	No specific interventions. Consider laser therapy for aesthetic concerns.

Abbreviations: mIAS, mammalian target of rapamycin (mTOR) inhibitor associated stomatitis; irAEs, immunotherapy-related adverse events; VEGF, vascular endothelial growth factor; HhSP, hedgehog signaling pathway.

## Data Availability

Not applicable.

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
