# Peer review of "Pathogenesis of Oral Toxicities Associated with Targeted Therapy and Immunotherapy"

_ijms, 2023, doi:10.3390/ijms24098188_

Round 1
Reviewer 1 Report
Dear authors,
Concerning your article entitled "Pathogenesis of Oral Toxicities Associated with Targeted and Immunotherapy", I would like to begin by commending you on your work. The article provides a comprehensive overview of the subject matter, and it is evident that significant effort has been made to cover a broad range of topics related to the pathogenesis of oral toxicities.
However, some of the pathogenesis descriptions provided in the article were brief. Given the complexity of the mechanisms underlying oral toxicities associated with targeted and immunotherapy, I believe that a more detailed explanation of the pathogenesis would be beneficial for readers.
1. Title.
Good.
2. Abstract.
Good.
3. Introduction.
Reference styling is not in accordance of that with MDPI. (MDP à [1], [2] …).
4. Oral toxicities
· The absence of any illustration about the pathophysiology of the described toxicities made it hard to follow and concentrate on the text.
· The treatments were mentioned very briefly. Authors should explain more the available treatment options although that this is not the main aim of the manuscript.
· Line 166. “. A recent meta-analysis showed that although dysgeusia 166 had been reported in patient receiving ICI therapy, the risk was lower compared with 167 conventional chemotherapy regimens 38. The damage to peripheral nerves in neurologic ir- 168 AEs appears to be secondary to antibody responses or cell-mediated mechanisms to 169 compact myelin, Schwann cells or nodal antigens as part of an abnormal immune response 170 39. In addition, cross-reactivity between the tumor antigens and similar epitopes on healthy 171 cells have been reported as another possible mechanism of neurologic toxicity of ICIs 40”.this paragraph should be in the dysgeusia section.
· Dysgeusia section.
Overall, the paragraph appears to be well-written and informative. It provides relevant information about dysgeusia as a common adverse effect of cancer treatment and how it can be explained by neurological or mucosal damage. The paragraph also discusses how targeted therapies can induce dysgeusia and highlights the example of HhSP inhibitors used in the treatment of advanced, metastatic, or unresectable basal cell carcinoma. One suggestion for improvement would be to provide more detail on the underlying mechanisms of dysgeusia induced by HhSP inhibitors. The paragraph briefly mentions that HhSP regulates differentiation and maintenance of lingual taste receptor cells, but it would be helpful to further explain how this leads to taste alterations. Another area for improvement is the use of abbreviations. While some abbreviations, such as ALK and TKI, are commonly used in the field, others, such as HhSP, may be less familiar to readers. It would be helpful to introduce the full name of the abbreviation upon first use in the text. Overall, the paragraph provides useful information on dysgeusia as a common toxicity of cancer treatment and its mechanisms in the context of targeted therapies.
· Lichenoid Lesions.
Overall, this paragraph provides a good overview of lichenoid lesions as an adverse event of imatinib treatment. It is helpful that:
o the paragraph specifies the frequency of oral lichenoid reactions associated with imatinib use and the characteristic features of these lesions.
o The inclusion of the management options for symptomatic cases is also valuable. However, it would be useful to have more information on the pathogenesis of lichenoid lesions associated with imatinib and rituximab, as well as any potential risk factors or patient populations that may be more susceptible to developing these lesions.
o It would be beneficial to provide more specific details regarding the use of systemic therapies in severe cases, such as what types of medications are typically used and their effectiveness.
· Pigmentation section.
Pigmentations are benign; however, I do not agree that pigmentations do not require treatment. Pigmentation requires treatments for aesthetic aims. In this context, periodontists and/or oral surgeon uses methods such as laser depigmentation.
· Geographic tongue
It is true that geographic tongue is often asymptomatic but can be associated with pain and/or a burning sensation. It would be helpful to provide some statistics or more specific information on how often this occurs, so that readers can better understand the potential impact of geographic tongue.
Few treatment options for symptomatic cases were described. More information on how effective these treatments are, as well as any potential side effects or drawbacks must be added.
Nothing related to the pathogenesis of geographic tongue and the targeted and immunotherapy was mentioned. Although not clearly understood, hypothesis must be mentioned in this paragraph.
5. GENERALLY
· Other than that, the manuscript is well written and gives an overall review on the toxicities related to targeted therapy and immunotherapy.
· In literature there is a lack of studies that review Pathogenesis of oral toxicities associated with targeted and immunotherapy.
· Figures illustrating some of the complicated pathogenesis of the oral toxicities will help very much the readers.
Author Response
We are thankful for the reviewers’ thoughtful comments. We have considered them carefully and revised the manuscript accordingly. Our point-by-point responses are given below. Please find enclosed the revised manuscript with “track changes”. Please, let me know if you have any questions.
Thank you again for your support.
REVIEWER 1
Dear authors,
Concerning your article entitled "Pathogenesis of Oral Toxicities Associated with Targeted and Immunotherapy", I would like to begin by commending you on your work. The article provides a comprehensive overview of the subject matter, and it is evident that significant effort has been made to cover a broad range of topics related to the pathogenesis of oral toxicities.
RESPONSE: thanks for the encouraging words. We are grateful for your suggestions and made several changes based on your recommendations below.
However, some of the pathogenesis descriptions provided in the article were brief. Given the complexity of the mechanisms underlying oral toxicities associated with targeted and immunotherapy, I believe that a more detailed explanation of the pathogenesis would be beneficial for readers.
- Title.
Good.
- Abstract.
Good.
- Introduction.
Reference styling is not in accordance of that with MDPI. (MDP à [1], [2] …).
RESPONSE: thanks for your comment. We were told by the editorial office that all references will be reformatted by the journal staff if the manuscript is accepted.
- Oral toxicities
- The absence of any illustration about the pathophysiology of the described toxicities made it hard to follow and concentrate on the text.
RESPONSE: Thanks for the suggestions. We now included a Table (Table 1) to summarize the main findings by type of oral toxicity.
- The treatments were mentioned very briefly. Authors should explain more the available treatment options although that this is not the main aim of the manuscript.
RESPONSE: We have expanded on the treatment options available where possible. However, some of the oral toxicities were reported in case reports or small series, and it can be challenging to draw any specific conclusions.
- Line 166. “. A recent meta-analysis showed that although dysgeusia 166 had been reported in patient receiving ICI therapy, the risk was lower compared with 167 conventional chemotherapy regimens 38. The damage to peripheral nerves in neurologic ir- 168 AEs appears to be secondary to antibody responses or cell-mediated mechanisms to 169 compact myelin, Schwann cells or nodal antigens as part of an abnormal immune response 170 39. In addition, cross-reactivity between the tumor antigens and similar epitopes on healthy 171 cells have been reported as another possible mechanism of neurologic toxicity of ICIs 40”.this paragraph should be in the dysgeusia section.
RESPONSE: Thank you for your feedback. We have made changes to the manuscript based on suggestions from other reviewers and moved this paragraph to the dysgeusia section.
Dysgeusia section. Overall, the paragraph appears to be well-written and informative. It provides relevant information about dysgeusia as a common adverse effect of cancer treatment and how it can be explained by neurological or mucosal damage. The paragraph also discusses how targeted therapies can induce dysgeusia and highlights the example of HhSP inhibitors used in the treatment of advanced, metastatic, or unresectable basal cell carcinoma. One suggestion for improvement would be to provide more detail on the underlying mechanisms of dysgeusia induced by HhSP inhibitors. The paragraph briefly mentions that HhSP regulates differentiation and maintenance of lingual taste receptor cells, but it would be helpful to further explain how this leads to taste alterations.
RESPONSE: HhSP plays an important role in regulating the differentiation and maintenance of lingual taste receptor cells. Disruptions to this pathway can lead to alterations in the development and function of these cells, which may result in taste alterations. This has been shown through various methods such as activation, inhibition, suppression, and genetic deletion of key components, all of which led to alterations in taste buds and loss of taste nerve responses to chemical stimuli. We modified the paragraph with the addition of the following text: “Kumari et al (2015) investigated the role of the Hh pathway in taste sensation by using sonidegib (LDE225) to inhibit signaling at Smoothened, a key regulator of Hh signaling. Normally, when the Hh ligand binds to the membrane receptor Patched, Smoothened is activated and initiates a signaling cascade that leads to the activation of Gli transcription factors. Treatment with sonidegib demonstrated a direct and essential requirement of the Hh pathway in maintaining taste bud function and homeostasis. Furthermore, the results of this study, along with other studies using the HPI drug sonidegib, showed that disrupting Hh signaling led to rapid loss of taste buds and reduced levels of Shh ligand within the taste bud, indicating loss of taste bud cells.”
Another area for improvement is the use of abbreviations. While some abbreviations, such as ALK and TKI, are commonly used in the field, others, such as HhSP, may be less familiar to readers. It would be helpful to introduce the full name of the abbreviation upon first use in the text.
RESPONSE: Thanks for your comment. We introduced the full name for all the abbreviations as recommended by the reviewer.
Lichenoid Lesions. Overall, this paragraph provides a good overview of lichenoid lesions as an adverse event of imatinib treatment. It is helpful that:
- the paragraph specifies the frequency of oral lichenoid reactions associated with imatinib use and the characteristic features of these lesions.
RESPONSE: We thank the reviewer for the comment and suggestion. We included the characteristic features in paragraph 2.2.2 (“The lichenoid lesions are typically asymptomatic, presenting with characteristic reticular, erosive, and/or ulcerative features”).
- The inclusion of the management options for symptomatic cases is also valuable. However, it would be useful to have more information on the pathogenesis of lichenoid lesions associated with imatinib and rituximab, as well as any potential risk factors or patient populations that may be more susceptible to developing these lesions.
RESPONSE: Thanks for the comment. We included the following section as suggested: “The exact mechanism by which imatinib causes oral mucosal pigmentation remains unclear. Imatinib is known to target the ATP-binding site of the Bcr-Abl tyrosine kinase, as well as other tyrosine kinases like platelet-derived growth factor receptor-b, C-kit, and C-ABL. C-kit is a transmembrane growth factor expressed in melanocytes, basal skin cells, and mast cells. Stimulation of C-kit leads to the activation of the microphthalmia transcription factor (MITF), which then transactivates the promoter of the tyrosinase pigmentation gene of melanocytes. Imatinib is thought to inhibit ligand binding to specific receptors on the surface of human melanocytes, which reduces cellular activity and may cause hypopigmentation and rarely cause hyperpigmentation of the skin and mucosa. This may be due to a metabolite of the drug chelating iron and melanin, similar to the action of other drugs (e.g., minocycline and anti-malarial drugs). The reason why the hard palatal mucosa is often affected is not yet understood, but it is known that the palatal mucosa contains a large number of mucosal melanocytes where imatinib metabolites can accumulate. Additionally, C-kit signalling may also play a role in oral hyperpigmentation as it is expressed in mesenchymal cells of the human oral cavity such as dental pulp cells and gingival fibroblasts.”
o It would be beneficial to provide more specific details regarding the use of systemic therapies in severe cases, such as what types of medications are typically used and their effectiveness.
RESPONSE: thanks for the suggestion. We expanded on the treatment options available and included the following sentence: “The primary objective of the treatment is to alleviate discomfort, as well as reducing or resolving erythema and ulcers. In some cases, systemic corticosteroids, as well as other immunosuppressive drugs may be required.”
Pigmentation section. Pigmentations are benign; however, I do not agree that pigmentations do not require treatment. Pigmentation requires treatments for aesthetic aims. In this context, periodontists and/or oral surgeon uses methods such as laser depigmentation.
RESPONSE: We agree with the reviewer that pigmentations may require treatment if aesthetic is a concern. However, there’s no literature on the use of laser treatment for oral cavity pigmentations secondary to targeted agents. We included the following sentence as suggested “Laser therapy may be recommended for individuals with extensive pigmentations and or aesthetic concerns.”
- Geographic tongue. It is true that geographic tongue is often asymptomatic but can be associated with pain and/or a burning sensation. It would be helpful to provide some statistics or more specific information on how often this occurs, so that readers can better understand the potential impact of geographic tongue. Few treatment options for symptomatic cases were described. More information on how effective these treatments are, as well as any potential side effects or drawbacks must be added.
RESPONSE: Thanks for the suggestions. Unfortunately, it is challenging to determine the precise number of cases of symptomatic geographic tongue since numerous instances may remain unreported or undiagnosed. We added the following sentence to section 2.2.5: “it is difficult to estimate the exact number of cases of symptomatic geographic tongue, as many cases may go unreported or undiagnosed.”
Nothing related to the pathogenesis of geographic tongue and the targeted and immunotherapy was mentioned. Although not clearly understood, hypothesis must be mentioned in this paragraph.
RESPONSE: We expanded on this and added the following section: “The pathogenesis of geographic tongue secondary to targeted therapy is not well understood. Hubiche et al suggested that since VEGF or VEGF receptors play a crucial role in maintaining the homeostasis of the buccal mucosa, as well as in certain oral diseases angiogenesis inhibitors for targeting these specific molecules could potentially induce the development of geographic tongue”
GENERALLY
- Other than that, the manuscript is well written and gives an overall review on the toxicities related to targetedtherapy and immunotherapy.
RESPONSE: thank you for the helpful suggestions and for taking the time to review our work.
- In literature there is a lack of studies that review Pathogenesis of oral toxicities associated with targeted and immunotherapy.
RESPONSE: the reviewer makes a great point. One of the major limitations for oral toxicities from targeted therapies and immunotherapy is that very few studies examined the pathobiology of these complications. Most literature focuses on the clinical description of the toxicities (except for conventional oral mucositis)
- Figures illustrating some of the complicated pathogenesis of the oral toxicities will help very much the readers.
RESPONSE: Thanks for the suggestions. Since the pathobiology of many complications is still not fully understood we believe a figure for each toxicity would not be a feasible approach. However, we have included a new table (Table 1) to summarize the main findings by type of oral toxicity.
Reviewer 2 Report
This is an interesting narrative review with valuable clinical information.
Minor consideration
The categorization of oral toxicities is not consistent: the first two categories are by the class of drugs (mTOR, ICI), but the following are by symptom/sign (dysesthesia, dysgeusia). I suggest that the author should use the same classification criteria throughout the manuscript and change the titles/content of sub-heading accordingly. If they decide to present toxicities by class of drug, then they should follow the order that the drugs appear in the third paragraph of the Introduction (or change their order there) for consistency.
Minor consideration
I suggest that the authors should carefully re-read their manuscript, as there are many “minor” errors that hamper the quality of their work. Some examples:
L46 and L130: “organs, including the oral cavity”. Please rephrase, oral cavity is an anatomical location, not an organ.
L52: “differs from the one”. Differ from the ones?
L62: “xerostomia, and salivary gland hypofunction”. Please delete “,”.
L78: “Mammalian target of rapamycin inhibitors (mTORi)”. The acronym has been previously defined, you can simply use mTORi.
L79: “of several malignancies of the pancreas, lung, and advanced breast”. Please rephrase. What is the meaning of “advanced breast”? In the same sentence, use “;” or “,” to separate phrases, not both of them.
L83: “currently mTORi approved”. mTORi currently approved by…
L97-922: “In addition to mIAS several cases of dysgeusia and xerostomia have been reported in patients receiving…”. This is an example of how “variable categorization” affects the content: this sentence should better appear in the respective section for xerostomia/dysgeusia.
L110-118: The information appearing in those lines has no reference(s). Is it from the study of Sonis et al [21]?
L126-128: “ICIs target programmed cell death-1 (PD-1)…”. This sentence is inconclusive.
L132 Immune related adverse events. Please use the same acronym throughout the manuscript (appears as ir-AEs, ir-Aes, irAes).
L135: “xerostomia, are”. Please delete “,”.
L156: ”Sjögren disease-like”. It is Sjogren syndrome. Please correct appropriately in other lines of the manuscript, as well.
L162: “regulat dental”. Please correct “regular”.
L190: “to what it is recommended for patients with burning mouth syndrome; and involves the use”. Please rephrase, “;” is not proper here.
L208 “per NCI-CTCAE” and L215 “MET, and ROS1”. Please explain the acronyms. I suggest that you check for similar issues throughout the manuscript.
L261” in a patient treated vemurafenib monotherapy”. Please rephrase.
L277: “Of note, the existing literature does not clearly establish whether patients developed geographic tongue as a result of targeted therapy or if the condition worsened following administration of the targeted agents”. Is there the chance that the condition was noticed after the commencement of the drug, as an incidental finding, but was present for years before?
L298: “Anecdotal cases of intraoral hyperpigmentation affecting other sites such as the gingivae or teeth have also been reported.” Please give reference.
Author Response
We are thankful for the reviewer’s thoughtful comments. We have considered them carefully and revised the manuscript accordingly. Our point-by-point responses are given below. Please find enclosed the revised manuscript with “track changes”. Please, let me know if you have any questions.
Thank you again for your support.
REVIEWER 2
This is an interesting narrative review with valuable clinical information.
RESPONSE: we thank the reviewer for the encouraging comment.
Minor consideration
The categorization of oral toxicities is not consistent: the first two categories are by the class of drugs (mTOR, ICI), but the following are by symptom/sign (dysesthesia, dysgeusia). I suggest that the author should use the same classification criteria throughout the manuscript and change the titles/content of sub-heading accordingly. If they decide to present toxicities by class of drug, then they should follow the order that the drugs appear in the third paragraph of the Introduction (or change their order there) for consistency.
RESPONSE: Thanks for your comment. We now used the same classification criteria throughout the manuscript as suggested (Stomatitis, Red and white lesions, Oral dysesthesia, Dysgeusia, Salivary gland hypofunction, Gingival bleeding, and Hyperpigmentation)
Minor consideration
I suggest that the authors should carefully re-read their manuscript, as there are many “minor” errors that hamper the quality of their work. Some examples:
RESPONSE: thank you for your suggestions below. We carefully proofread the manuscript and made the necessary changes.
L46 and L130: “organs, including the oral cavity”. Please rephrase, oral cavity is an anatomical location, not an organ.
RESPONSE: Thanks for noticing this. We changed it to “sites.”
L52: “differs from the one”. Differ from the ones?
RESPONSE: the suggested change was made.
L62: “xerostomia, and salivary gland hypofunction”. Please delete “,”.
RESPONSE: the suggested edit was made.
L78: “Mammalian target of rapamycin inhibitors (mTORi)”. The acronym has been previously defined, you can simply use mTORi.
RESPONSE: the suggested revision was made.
L79: “of several malignancies of the pancreas, lung, and advanced breast”. Please rephrase. What is the meaning of “advanced breast”? In the same sentence, use “;” or “,” to separate phrases, not both of them.
RESPONSE: we added the word “cancer’ after breast. We used “;” only in the sentence as recommended.
L83: “currently mTORi approved”. mTORi currently approved by…
RESPONSE: the suggested revision was made.
L97-922: “In addition to mIAS several cases of dysgeusia and xerostomia have been reported in patients receiving…”. This is an example of how “variable categorization” affects the content: this sentence should better appear in the respective section for xerostomia/dysgeusia.
RESPONSE: as noted in our response above we now used the same classification criteria throughout the manuscript as suggested (Stomatitis, Red and white lesions, Oral dysesthesia, Dysgeusia, Salivary gland hypofunction, Gingival bleeding, and Hyperpigmentation)
L110-118: The information appearing in those lines has no reference(s). Is it from the study of Sonis et al [21]?
RESPONSE: Yes. We changed the text clarified this in the revised version of the manuscript (“In particular, in their organotypic model, everolimus was shown to inhibit the prolifer-ation of epithelial cells and induce apoptotic changes with the formation of intra-epithelial vacuole.”.
L126-128: “ICIs target programmed cell death-1 (PD-1)…”. This sentence is inconclusive.
RESPONSE: we checked the sentence and made a minor edit. It now reads “ICIs target programmed cell death-1 (PD-1), PD ligand-1 (PD-L1) and cytotoxic T-lymphocyte-associated antigen-4 (CTLA-4) with a subsequent disruption of the host T cell signaling, and the upregulation of the T cell immune innate and adaptive response against cancer cells”
L132 Immune related adverse events. Please use the same acronym throughout the manuscript (appears as ir-AEs, ir-Aes, irAes).
RESPONSE: the suggested change was made. Thank you.
L135: “xerostomia, are”. Please delete “,”.
RESPONSE: the suggested revision was made.
L156: ”Sjögren disease-like”. It is Sjogren syndrome. Please correct appropriately in other lines of the manuscript, as well.
RESPONSE: the suggested revision was made.
L162: “regulat dental”. Please correct “regular”.
RESPONSE: we corrected the spelling as suggested. Thank you!
L190: “to what it is recommended for patients with burning mouth syndrome; and involves the use”. Please rephrase, “;” is not proper here.
RESPONSE: Thanks for the comment. We changed it to “Management focuses on relieving symptoms and is similar to the recommended approach for patients with burning mouth syndrome. It involves using topical benzodiazepines or systemic anti-convulsant drugs, tricyclic anti-depressants, with varying responses.”
L208 “per NCI-CTCAE” and L215 “MET, and ROS1”. Please explain the acronyms. I suggest that you check for similar issues throughout the manuscript.
RESPONSE: we explained the acronyms in the text as recommended. Please know that ROS1 is the actual name of a proto oncogene.
L261” in a patient treated vemurafenib monotherapy”. Please rephrase.
RESPONSE: Thanks. We rephrased the sentence as suggested (“While primarily involving the skin, oral mucosal hyperkeratosis has also been reported to occur within the first weeks of treatment, as well as a single report of SCC of the labial mucosa developing in a patient treated with vemurafenib”).
L277: “Of note, the existing literature does not clearly establish whether patients developed geographic tongue as a result of targeted therapy or if the condition worsened following administration of the targeted agents”. Is there the chance that the condition was noticed after the commencement of the drug, as an incidental finding, but was present for years before?
RESPONSE: thanks for bringing this up. It is definitely a possibility and several studies do not include information on this. We added this point in the section on geographic tongue.
L298: “Anecdotal cases of intraoral hyperpigmentation affecting other sites such as the gingivae or teeth have also been reported.” Please give reference
RESPONSE: We added two references on dental and gingival pigmentations as recommended.
Reviewer 3 Report
This manuscript is a well-written paper that reviewed oral complications after targeted and immunotherapy classified into the following 10 categories; mammalian target of rapamycin inhibitors associated stomatitis, oral immune-related adverse events, oral dysesthesia, dysgeusia, lichenoid lesions, hyperkeratotic lesions and possible increased risk of squamous cell carcinoma, geographic tongue, gingival bleeding, hyperpigmentation, stomatitis secondary to anti-EGFR agents and VEGFR inhibitors. However, subheadings include both symptomatic and causative classifications, which makes the context complex. For example, the following subheadings are suitable to simplify the context; 1)stomatitis, 2) dry mouth and xerostomia, 3) oral dysesthesia, 4) dysgeusia, 5) lichenoid lesion and other mucosal lesions, 6) hyperkeratotic lesions and possible increased risk of squamous cell carcinoma, 7) geographic tongue, 8) gingival bleeding, and 9) hyperpigmentation. Furthermore, I recommend summarizing your results in a table with symptoms, associated drugs, and treatments. Others- This abstract stated only the aim of this review report. Please briefly describe results and your conclusion.
- Frequent use of unusual abbreviations, e.g., ir-Aes might make readers fatigued.
Author Response
We are thankful for the reviewer’s thoughtful comments. We have considered them carefully and revised the manuscript accordingly. Our point-by-point responses are given below. Please find enclosed the revised manuscript with “track changes”. Please, let me know if you have any questions.
Thank you again for your support.
REVIEWER 3
This manuscript is a well-written paper that reviewed oral complications after targeted and immunotherapy classified into the following 10 categories; mammalian target of rapamycin inhibitors associated stomatitis, oral immune-related adverse events, oral dysesthesia, dysgeusia, lichenoid lesions, hyperkeratotic lesions and possible increased risk of squamous cell carcinoma, geographic tongue, gingival bleeding, hyperpigmentation, stomatitis secondary to anti-EGFR agents and VEGFR inhibitors. However, subheadings include both symptomatic and causative classifications, which makes the context complex. For example, the following subheadings are suitable to simplify the context; 1)stomatitis, 2) dry mouth and xerostomia, 3) oral dysesthesia, 4) dysgeusia, 5) lichenoid lesion and other mucosal lesions, 6) hyperkeratotic lesions and possible increased risk of squamous cell carcinoma, 7) geographic tongue, 8) gingival bleeding, and 9) hyperpigmentation.
RESPONSE: We thank the reviewer for the encouraging words. We have modified the outline as suggested, now using the same classification criteria throughout the manuscript: 1) Stomatitis, 2) Red and white lesions, 3) Oral dysesthesia, 4) Dysgeusia, 5) Salivary gland hypofunction, 6) Gingival bleeding, and 7) Hyperpigmentation.
Furthermore, I recommend summarizing your results in a table with symptoms, associated drugs, and treatments.
RESPONSE: Thank you for this valuable suggestion. We have added “Table 1. Oral Toxicities associated with Targeted and Immunotherapies: Summary Table" accordingly.
Oral Toxicities |
Drug |
Class |
Main Treatment Suggestions |
Stomatitis,
mIAS |
Cetuximab, panitumumab, trastuzumab; gefitinib, erlotinib, afatinib, dacomitinib
Sunitinib, sorafenib, pazopanib, cabozantinib
Everolimus, sirolimus, temsirolimus |
EGFR inhibitors (monoclonal antibodies; small-molecule tyrosine kinase inhibitors, respectively)
Multitargeted tyrosine kinase inhibitors
mTOR inhibitors |
Basic oral care, steroids (topical, intralesional, oral), analgesics |
Lichenoid lesions,
Oral mucosal irAEs |
Imatinib
Rituximab
Nivolumab, pembrolizumab, cemiplimab, ipilimumab, atezolizumab |
BCR-ABL inhibitor
Anti-CD20 monoclonal antibody
Immune checkpoint inhibitors |
Steroids (topical, intralesional, oral) |
Hyperkeratotic lesions |
Vemurafenib, dabrafenib, encorafenib |
BRAF inhibitors |
No specific interventions, routine examination and biopsy in case of irregular lesions |
Geographic tongue |
Bevacizumab
Sunitinib, sorafenib, axitinib |
Anti-VEGF monoclonal antibody
Multitargeted tyrosine kinase inhibitors |
Topical steroids for symptomatic cases |
Dysesthesia |
Sunitinib, sorafenib, pazopanib, cabozantinib
|
Multitargeted tyrosine kinase inhibitors
|
Topical or oral clonazepam, gabapentin, antidepressants. |
Dysgeusia |
Vismodegib, sonidegib
Crizotinib
Nivolumab, pembrolizumab, cemiplimab, ipilimumab, atezolizumab
Everolimus, temsirolimus |
HhSP inhibitors
Multitargeted tyrosine kinase inhibitor
Immune checkpoint inhibitors
mTOR inhibitors |
Consider consultation with dietitian. Taste usually returns after discontinuation of medication. |
Immune related salivary gland hypofunction, xerostomia |
Nivolumab, pembrolizumab, cemiplimab, ipilimumab, atezolizumab |
Immune checkpoint inhibitors
|
Basic oral care, topical mucosal lubricants, saliva substitutes, masticatory stimulants (sugar-free chewing gum or lozenges), sialogogues (oral pilocarpine, civemiline). |
Gingival bleeding |
Bevacizumab
Sunitinib, sorafenib |
Anti-VEGF monoclonal antibody
Multitargeted tyrosine kinase inhibitors |
No specific interventions. Maintain oral hygiene |
Hyperpigmentation |
Imatinib
|
BCR-ABL inhibitor
|
No specific interventions. Consider laser therapy for aesthetic concerns. |
Abbreviations: mIAS, mammalian target of rapamycin (mTOR) inhibitor associated stomatitis; irAEs, immune therapy related adverse events; VEGF, vascular endothelial growth factor; HhSP, hedgehog signaling pathway.
Others
- This abstract stated only the aim of this review report. Please briefly describe results and your conclusion.
RESPONSE: We have expanded the abstract as suggest. It now reads: “Targeted therapy and immunotherapy have redefined cancer treatment. While enhancing tumor response and improving survival rates in many cancer types, toxicities continue to occur, and often involve the oral cavity. Broadly reported as ‘mucositis’ or ‘stomatitis,’ oral toxicities induced by targeted therapies differ clinically and mechanistically from those associated with conventional chemotherapy. Manifesting primarily as mucosal lesions, salivary gland hypofunction, or orofacial neuropathies, these oral toxicities may nonetheless lead to significant morbidity and impact patients’ quality of life, thereby compromising clinical outcomes. We conclude that familiarity with the spectrum of associated toxicities and understanding of their pathogenesis represent an important area of clinical research, and may lead to better characterization, prevention, and management of these adverse events.”
- Frequent use of unusual abbreviations, e.g., ir-Aes might make readers fatigued.
RESPONSE: The suggested change was made. Thank you.
Reviewer 4 Report
Comments to Author
This review mainly focused on oral toxicities of targeted and immunotherapy. The range of discussion is narrow, and most of the oral toxicities related with targeted and immunotherapy have been illustrated in the review. However, the content is lack of innovation, and there are many details needed to be emphasized.
1. The innovation of the review is not sufficient and the most of content have been summarized in other reviews, for example PMID: 34714553, 28224235, 34265157.
2. The content in part 2 oral toxicities seems to be disordered. For example, the stomatitis is both mentioned in 2.1 and 2.10. And “2.2 Oral immune-related adverse events” is not on the same level with other subtitles.
3. There is not any summative tables or figures in the review which might impede readers to understand the content of the review.
4. Some recommendation about how to reduce those oral toxicities should be added in the review.
5. There are a number of grammatical errors in the review, for example, in page 1, line 16, “Targeted therapy and immunotherapy have redefined cancer treatment, leading to enhanced tumor response and improved survival rates in many cancer types.” The same mistakes also happen in page 1, line 46. Please read carefully and refine sentences.
6. Refer22 maybe not be a suitable reference. Replace the more suitable one.
7. In page 3, line 126, “PD ligand 1” should be replaced with “PD ligand-1”. In page 4, line 148, “PD/PL-1 inhibitors” is not the correct expression. In page 4, line 152, “TNF alpha” is usually written as “TNF- alpha” or “TNF-α”. Pay more attention to the standard writing in the review.
8. “Interestingly, the use of TNF alpha and IL-6 inhibitors have been recommended for the management of several irAEs, suggesting that cytokine levels (general or tissue specific) may play a role in the pathogenesis of irAEs as well.” Relevant references need to be cited here and paraphrase in detail.
9. “However, most cases of dry mouth secondary to ICI therapy seem to mediated mainly by autoreactive T cells rather than the B cells typical of Sjögren disease”. The mechanism of dry mouth mediated by autoreactive T cells need to be discussed in detail. In page 7, line 170-172, the mechanism also needs to be supplied.
10. Many abbreviations appear but do not have original names or explanations. e.g. What is MET/ ROS1?
11. Refer65 is not accurate. Find a more suitable article to replace it, for example PMID: 28129674. There are the same problems in the whole article, please refine it carefully.
12. “Patients treated with antiangiogenic targeted therapies, VEGF- and VEGFR-inhibitors, are also at increased risk of bleeding and delayed wound healing because of the effect on vascular permeability and proliferation.” Explain the signaling pathway of vascular permeability and proliferation in detail.
13. In page 7, line 304, there is no suitable reference to explain the mechanism about “c-kit”.
14. In page 8, line 330, “Management of oral mucosal lesions” includes what kinds of therapies, paraphrase it.
Author Response
We are thankful for the reviewer’s thoughtful comments. We have considered them carefully and revised the manuscript accordingly. Our point-by-point responses are given below. Please find enclosed the revised manuscript with “track changes”. Please, let me know if you have any questions.
Thank you again for your support.
REVIEWER 4
This review mainly focused on oral toxicities of targeted and immunotherapy. The range of discussion is narrow, and most of the oral toxicities related with targeted and immunotherapy have been illustrated in the review. However, the content is lack of innovation, and there are many details needed to be emphasized.
- The innovation of the review is not sufficient and the most of content have been summarized in other reviews, for example PMID: 34714553, 28224235, 34265157.
RESPONSE: The authors acknowledge that previous reviews have addressed a similar topic. This is an invited review, and the topic was assigned to us by the guest editor. As such, the scope of work was somewhat limited. We appreciate the reviewer's feedback and have included additional information on toxicities that were not extensively discussed in the previously mentioned manuscripts. Furthermore, we have taken the reviewer's suggestions and expanded the discussion section.
The content in part 2 oral toxicities seems to be disordered. For example, the stomatitis is both mentioned in 2.1 and 2.10. And “2.2 Oral immune-related adverse events” is not on the same level with other subtitles.
RESPONSE: We have modified the outline as suggested, now using the same classification criteria throughout the manuscript: 1) Stomatitis, 2) Red and white lesions, 3) Oral dysesthesia, 4) Dysgeusia, 5) Salivary gland hypofunction, 6) Gingival bleeding, and 7) Hyperpigmentation.
- There is not any summative tables or figures in the review which might impede readers to understand the content of the review.
RESPONSE: We appreciate this suggestion and have added “Table 1. Oral Toxicities associated with Targeted and Immunotherapies: Summary Table" accordingly.
- Some recommendation about how to reduce those oral toxicities should be added in the review.
RESPONSE: The management of targeted therapy and immunotherapy-induced oral toxicities is generally focused on symptom control and is not prophylactic in nature. Largely, interventions are based on expert opinion. We highlight the main treatment suggestions in text and in the newly added Table 1. Our conclusion also holds a forward-facing statement in this regard, “A better understanding of the pathogenesis of oral toxicities from targeted therapy may lead to the development of reliable biomarkers for AE prediction and monitoring, new prevention strategies, and therapeutic options in the future.”
- There are a number of grammatical errors in the review, for example, in page 1, line 16, “Targeted therapy and immunotherapy have redefined cancer treatment, leading to enhanced tumor response and improved survival rates in many cancer types.” The same mistakes also happen in page 1, line 46. Please read carefully and refine sentences.
RESPONSE: Thank you for this comment. We have refined both sentences: “Targeted therapy and immunotherapy have redefined cancer treatment. While enhancing tumor response and improving survival rates in many cancer types, toxicities continue to occur, and often involve the oral cavity” and “The benefit from targeted therapy and immune checkpoint inhibitors is however tempered by toxicities that affect different sites, including the oral cavity.”
- Refer22 maybe not be a suitable reference. Replace the more suitable one.
RESPONSE: Thanks for catching that. We included a new reference for this sentence.
- In page 3, line 126, “PD ligand 1” should be replaced with “PD ligand-1”. In page 4, line 148, “PD/PL-1 inhibitors” is not the correct expression. In page 4, line 152, “TNF alpha” is usually written as “TNF- alpha” or “TNF-α”. Pay more attention to the standard writing in the review.
RESPONSE: The suggested changes were made. Thank you.
- “Interestingly, the use of TNF alpha and IL-6 inhibitors have been recommended for the management of several irAEs, suggesting that cytokine levels (general or tissue specific) may play a role in the pathogenesis of irAEs as well.” Relevant references need to be cited here and paraphrase in detail.
RESPONSE: We have added references as 1. The text now reads: “the use of TNF -α (e.g., infliximab) and IL-6 (e.g., tocilizumab) inhibitors as steroid-sparing agents has been recommended for the management of several irAEs, suggesting that cytokine levels (general or tissue specific) may play a role in the pathogenesis of irAEs as well. Indeed, both CTLA-4 and PD-1/PD- L1 inhibition result in increased cytokine production, including TNF, INF-γ, and IL-2, which can lead to further T cell proliferation and activation. Further study is required to elucidate the precise role these cytokines play in the development of irAEs.”
- “However, most cases of dry mouth secondary to ICI therapy seem to mediated mainly by autoreactive T cells rather than the B cells typical of Sjögren disease”. The mechanism of dry mouth mediated by autoreactive T cells need to be discussed in detail. In page 7, line 170-172, the mechanism also needs to be supplied.
RESPONSE: We have expanded on the mechanism of dry mouth as suggested. This section now reads: “Patients with salivary gland hypofunction may complain of severe dry mouth with Sjögren syndrome -like clinical features and symptoms. However, most cases of dry mouth secondary to ICI therapy seem to be mediated mainly by autoreactive T cells and T cell-mediated inflammation of salivary glands rather than the B cells typical of Sjögren syndrome. Indeed, labial salivary gland biopsies obtained from ICI-induced sicca patients demonstrated marked sialadenitis with increased CD3+T cell infiltration and acinar injury but a virtual absence of CD20+ B cells. Furthermore, only a few ICI- induced patients were seropositive for anti-Sjögren Syndrome-related Antigens A or B (Anti-SSA/B) autoantibodies, which may have been pre-existing. Management of immune therapy related xerostomia (“subjective feeling of oral dryness”) and dry mouth is with oral moisturizers or sialogogue therapy (pilocarpine and cevimeline) 2. With increased risk of dental sequelae (e.g., caries, recurrent candidiasis infections), regular dental examination including frequent dental prophylaxis and prescription topical fluoride treatments is recommended.”
- Many abbreviations appear but do not have original names or explanations. e.g. What is MET/ ROS1?
RESPONSE: We explained the acronyms in the text as recommended. Please note that ROS1 is the actual name of a proto-oncogene.
- Refer65 is not accurate. Find a more suitable article to replace it, for example PMID: 28129674. There are the same problems in the whole article, please refine it carefully.
RESPONSE: Thank you for your comment. We have added the reference Wu et al. 2017 to the section on hyperkeratotic lesions, as suggested. Reference #65 (now #23), Dermatologic Adverse Events of Systemic Anticancer Therapies: Cytotoxic Chemotherapy, Targeted Therapy, and Immunotherapy by Deutsch et al. (2020) is a recent, highly cited comprehensive review on this topic. It would be a miss if the authors were to remove it.
- “Patients treated with antiangiogenic targeted therapies, VEGF- and VEGFR-inhibitors, are also at increased risk of bleeding and delayed wound healing because of the effect on vascular permeability and proliferation.” Explain the signaling pathway of vascular permeability and proliferation in details
RESPONSE: We have expanded on the hypothesized mechanism leading to increased bleeding events as suggested: “Patients treated with antiangiogenic targeted therapies, VEGF- and VEGFR-inhibitors, are also at increased risk of bleeding and delayed wound healing be-cause of the effect on vascular permeability and proliferation. The precise mechanism by which bleeding is precipitated has not been full elucidated. The main hypothesis is that VEGF signaling plays a role in promoting endothelial cell survival and integrity in the adult vasculature, therefore, its inhibition interferes with the regenerative capacity of damaged endothelial cells and causes capillary leakage.”
- In page 7, line 304, there is no suitable reference to explain the mechanism about “c-kit”.
RESPONSE: Thank you for the comment. We have revised the text and added references as suggested: “The exact mechanism by which imatinib causes oral mucosal pigmentation remains unclear. Imatinib is known to target the ATP-binding site of the Bcr-Abl tyrosine kinase, as well as other tyrosine kinases like platelet-derived growth factor receptor-b, C-kit, and C-ABL. C-kit is a transmembrane growth factor expressed in melanocytes, basal skin cells, and mast cells. Stimulation of C-kit leads to the activation of the microphthalmia transcription factor (MITF), which then transactivates the promoter of the tyrosinase pigmentation gene of melanocytes. Imatinib is thought to inhibit ligand binding to specific receptors on the surface of human melanocytes, which reduces cellular activity and may cause hypopigmentation and rarely cause hyperpigmentation of the skin and mucosa. This may be due to a metabolite of the drug chelating iron and melanin, similar to the action of other drugs (e.g., minocycline and anti-malarial drugs). The reason why the hard palatal mucosa is often affected is not yet understood, but it is known that the palatal mucosa contains a large number of mucosal melanocytes where imatinib metabolites can accumulate. Additionally, C-kit signalling may also play a role in oral hyperpigmentation as it is expressed in mesenchymal cells of the human oral cavity such as dental pulp cells and gingival fibroblasts.”
- In page 8, line 330, “Management of oral mucosal lesions” includes what kinds of therapies, paraphrase it
RESPONSE: Thank you for your comment. We acknowledge that the wording was unclear and have revised the text to clarify the scope of suggested interventions: “Management of oral mucosal lesions caused by EGFRIs and multitargeted TKIs follows expert opinion recommendations for targeted therapy-associated stomatitis, as published by the European Society of Medical Oncology. Similar to the management of mIAS, in addition to basic oral care and oral hygiene recommendations, the use of high-potency steroids (topical, intralesional, or systemic) is recommended as first-line therapy.”